# The C-Terminal Domain of LRRK2 with the G2019S Substitution Increases Mutant A53T α-Synuclein Toxicity in Dopaminergic Neurons In Vivo

**DOI:** 10.3390/ijms22136760

**Published:** 2021-06-23

**Authors:** Noémie Cresto, Camille Gardier, Marie-Claude Gaillard, Francesco Gubinelli, Pauline Roost, Daniela Molina, Charlène Josephine, Noëlle Dufour, Gwenaëlle Auregan, Martine Guillermier, Suéva Bernier, Caroline Jan, Pauline Gipchtein, Philippe Hantraye, Marie-Christine Chartier-Harlin, Gilles Bonvento, Nadja Van Camp, Jean-Marc Taymans, Karine Cambon, Géraldine Liot, Alexis-Pierre Bemelmans, Emmanuel Brouillet

**Affiliations:** 1Université Paris-Saclay, CEA, CNRS, Laboratoire des Maladies Neurodégénératives, MIRCen, F-92265 Fontenay-aux-Roses, France; Crestonoemie@gmail.com (N.C.); Camille.gardier@gmail.com (C.G.); Marie-Claude.gaillard@cea.fr (M.-C.G.); Francescogubinelli@gmail.com (F.G.); pauline_roost@live.nl (P.R.); dfmolina@uc.cl (D.M.); Charlene.josephine@cea.fr (C.J.); Noelle.dufour@cea.fr (N.D.); Gwenaelle.auregan@cea.fr (G.A.); martine.guillermier@cea.fr (M.G.); sueva.bernier@cea.fr (S.B.); caroline.jan@cea.fr (C.J.); pauline.gipchtein@cea.fr (P.G.); philippe.hantraye@cea.fr (P.H.); gilles.bonvento@cea.fr (G.B.); nadja.van-camp@cea.fr (N.V.C.); karine.cambon@cea.fr (K.C.); Geraldine.Liot@cea.fr (G.L.); Alexis.bemelmans@cea.fr (A.-P.B.); 2University of Lille, Inserm, CHU Lille, U1172 - LilNCog-Lille Neuroscience and Cognition, F-59000 Lille, France; marie-christine.chartier-harlin@inserm.fr (M.-C.C.-H.); jean-marc.taymans@inserm.fr (J.-M.T.); 3Brain Biology and Chemistry, LiCEND, F-59000 Lille, France

**Keywords:** Parkinson’s disease, leucine-rich repeat kinase 2, α-synuclein, AAVs, cell-autonomous mechanisms

## Abstract

Alpha-synuclein (α-syn) and leucine-rich repeat kinase 2 (LRRK2) play crucial roles in Parkinson’s disease (PD). They may functionally interact to induce the degeneration of dopaminergic (DA) neurons via mechanisms that are not yet fully understood. We previously showed that the C-terminal portion of LRRK2 (ΔLRRK2) with the G2019S mutation (ΔLRRK2^G2019S^) was sufficient to induce neurodegeneration of DA neurons in vivo, suggesting that mutated LRRK2 induces neurotoxicity through mechanisms that are (i) independent of the N-terminal domains and (ii) “cell-autonomous”. Here, we explored whether ΔLRRK2^G2019S^ could modify α-syn toxicity through these two mechanisms. We used a co-transduction approach in rats with AAV vectors encoding ΔLRRK2^G2019S^ or its “dead” kinase form, ΔLRRK2^DK^, and human α-syn with the A53T mutation (AAV-α-syn^A53T^). Behavioral and histological evaluations were performed at 6- and 15-weeks post-injection. Results showed that neither form of ΔLRRK2 alone induced the degeneration of neurons at these post-injection time points. By contrast, injection of AAV-α-syn^A53T^ alone resulted in motor signs and degeneration of DA neurons. Co-injection of AAV-α-syn^A53T^ with AAV-ΔLRRK2^G2019S^ induced DA neuron degeneration that was significantly higher than that induced by AAV-α-syn^A53T^ alone or with AAV-ΔLRRK2^DK^. Thus, mutated α-syn neurotoxicity can be enhanced by the C-terminal domain of LRRK2^G2019^ alone^,^ through cell-autonomous mechanisms.

## 1. Introduction

Parkinson’s disease (PD) is a neurodegenerative disorder that affects approximately seven million people worldwide. Early in the course of the disease, the most obvious symptoms are movement-related, including shaking (resting tremor), rigidity, and slowness of movement [1,2]. The neuropathological hallmarks of PD are characterized by the progressive loss of dopaminergic (DA) neurons in the substantia nigra *pars compacta* (SNpc) and the presence of neuronal aggregates (Lewy bodies) and dystrophic Lewy neurites containing the protein α-synuclein (α-syn) [3]. There is currently no treatment to delay such neurodegeneration, and the cause of α -syn aggregation and the preferential death of DA neurons is unknown. PD is mainly a sporadic neurodegenerative disorder, but approximately 10% of the cases are of genetic origin and several genes have been identified as causative factors [4].

Duplication, triplication, and rare mutations (A53T, A30P, E46K, H50Q, G51D, and A53E) in the SNCA gene, encoding the α-syn protein, have been found in families with dominantly-inherited PD and are associated with early-onset forms, with an amplification of α -syn aggregation [4,5,6,7]. The A53T [8], A30P [9], and E46K [10] substitutions have been the most studied thus far. Compelling evidence shows that α -syn takes center stage in PD and plays a key role via various aggregated forms, including abnormally phosphorylated aggregates, which produce multiple cellular alterations, eventually leading to the death of DA neurons [11].

Mutations in leucine-rich repeat kinase 2 (LRRK2) are the most common genetic cause of both familial and sporadic PD [12,13]. There are also variants in the LRRK2 locus that are considered to be risk factors for developing PD [14,15]. The most prevalent mutation in LRRK2 is the G2019S substitution, accounting for 5% to 6% of familial PD and 1% to 2% of de novo genetic PD cases [16,17]. Cases of patients harboring the G2019S and other mutations are clinically indistinguishable from those of idiopathic PD, most often including the presence of Lewy bodies (LBs) [18,19]. Although G2019S patients show clinical manifestations similar to those of sporadic patients [20], several studies have shown subtle differences [21,22]. Several have reported the presence of LBs in symptomatic LRRK2 mutation carriers, with LRRK2 found in the LBs [13]. However, this is still a subject of debate, as other neuropathological studies have instead reported the absence of detectable LBs in a sub-population of PD patients with LRRK2 mutations [23].

The mechanisms underlying the neurotoxicity of LRRK2 mutations are only partially understood. It is generally accepted that the G2019S mutation increases LRRK2 kinase activity and that neurotoxicity originates from such increased activity [24,25]. Autophosphorylation of LRRK2, phosphorylation of LRRK2 by exogenous kinases, and phosphorylation of LRRK2 substrates are all key determinants of LRRK2 toxicity. Although the C-terminal part of LRRK2, containing the enzymatically active part of the protein, appears to be crucial in producing neurotoxicity (when the protein harbors the G2019S substitution), the N-terminal part of the protein (which contains the armadillo, ankyrin, and LRR domains) also plays an important role [26]. The N-terminal domain interacts with multiple protein partners. For example, LRRK2 interactions with 14-3-3 and RAB proteins have been much studied [27]. In particular, RAB10 has been identified as a bona fide substrate of LRRK2, with LRRK2^G2019S^ resulting in increased phosphorylation of RAB10 in cells [28,29]. RAB10 phosphorylation is also elevated in the brains of patients with sporadic forms of PD [30]. These observations are crucial but do not rule out a role for other signaling pathways/mechanisms independent of the N-terminal part of LRRK2. Indeed, we previously showed that injection of adeno-associated virus (AAV) encoding the C-terminal portion of LRRK2 with the G2019S mutation (ΔLRRK2^G2019S^) containing the GTPase, kinase domains, and WD40 was sufficient to induce the degeneration of DA neurons in the substantia nigra pars compacta (SNpc) within six months post-infection [31]. In the aforementioned study, we showed that this fragment, in contrast to the full-length protein, does not interact with RAB10, as expected.

The central role of α-syn in the pathogenesis of PD has led to the hypothesis of a functional and, possibly, physical interaction between LRRK2 and α-syn (for a review [20,32]). Indeed, LRRK2 toxicity may require the presence of α-syn and, conversely, the presence of variant/mutant LRRK2 may increase the risk and/or impact of α-synucleopathy in PD. It has often been hypothesized that inhibiting the enzymatic activity of mutant LRRK2 or wild-type (WT) LRRK2 could reduce the impact of α-syn. The therapeutic implication of this hypothesis is extremely important; the regulation of LRRK2 kinase activity could be theoretically beneficial in slowing disease progression, not only in individuals harboring LRRK2 mutations, but also in idiopathic PD. Recent studies support this possibility. A pharmacological blockade of the kinase activity of WT *lrrk2* or mutant LRRK2^G2019S^ in animal models protects against the overexpression of human α-syn toxicity [33,34,35] (for a review, [20]).

However, this has been challenged by studies in which LRRK2 inhibitors, although efficiently reducing its kinase activity, do not prevent α-syn toxicity [36]. It has also been suggested that the key parameter of LRRK2 toxicity may not inherently reside in the activity of its kinase domain but rather in the level of expression of the LRRK2 protein in the cell [37]. In addition, if LRRK2 indeed potentiates α-syn toxicity in PD patients, it may not be solely linked to cell-autonomous mechanisms in DA neurons, but may also result from complex interactions between DA neurons and other LRRK2-expressing cells that surround DA neurons, especially microglial cells, astrocytes, and cells of the immune system, which likely play a role [38,39,40]

Here, we address these questions by studying the effect of ΔLRRK2^G2091S^ on the neurotoxicity of human α-syn with the A53T mutation (α-syn^A53T^). We performed experiments using AAVs that lead to the overexpression of the various forms of ΔLRRK2 and human α-syn^A53T^, alone or in combination, in DA neurons of the SNpc in adult rats. Results from the quantitative histological evaluation and behavioral assessment at two different time points after AAV injection suggest the existence of a cell-autonomous interplay between α-syn^A53T^ and LRRK2^G2019^, although of relatively low amplitude.

## 2. Results

### 2.1. Determination of the Experimental Conditions to Detect Potential Synergy between AAV-α-Syn^A53T^ and AAV-ΔLRRK2^G2019S^ Toxicity

We investigated whether human LRRK2 can increase the toxicity of human α-syn in DA neurons through cell-autonomous mechanisms. We used serotype 6 AAV capsids, which allow preferential expression in neuronal cells and lead to a high percentage of cells transduced in the injected structure, without excessive diffusion into the surrounding tissue, such as, for example, serotype 9 [41,42]. In a previous study [31], we showed that the C-terminal portion of human LRRK2^G2019S^ (ΔLRRK2^G2019S^, aa 1330–2527) retains, at least in part, the biochemical properties of full-length LRRK2^G2019S^, including higher kinase activity than the WT fragment. In addition, we found that overexpression of the C-terminal portion of human ΔLRRK2^G2019S^ in the adult rat SNpc, using AAVs, produced partial (~30%) but significant loss of DA neurons at 25 weeks post-transduction, whereas overexpression of the WT form of LRRK2 (ΔLRRK2^WT^) was not toxic [31]. Here, we used a similar approach using a slightly larger fragment (aa 1283–2527) (Figure 1A) to integrate serine 1292, which is thought to play an important role in LRRK2 activity [43].

We studied the effects of AAV-ΔLRRK2^G2019S^, AAV-ΔLRRK2^WT^, and AAV-ΔLRRK2^DK^ alone at 15 weeks PI. We aimed to find a PI time point associated with no or mild degeneration where ΔLRRK2 is expressed alone to be able to detect potential synergy with α-syn^A53T^ in subsequent co-expression experiments. We injected 4 µL AAV solution in all cases. A final amount of 2.5 × 10^10^ Vg per site and per vector was used. Each AAV was injected unilaterally into the SNpc (2.5 × 10^10^ Vg). In addition to the three experimental groups, a control group received injections of vehicle (PBS/pluronic acid). We evaluated the effect of a given AAV at 15 weeks PI using the cylinder test, a sensitive motor test able to detect asymmetry of the forepaws [44]. There was no major left/right forepaw asymmetry in rats injected with vehicle, AAV-ΔLRRK2^WT^, or AAV-ΔLRRK2^G2019S^ (Figure 1B). The AAV-ΔLRRK2^DK^ induced a statistically significant asymmetry (Figure 1B).

We assessed the integrity of the nigrostriatal pathway using unbiased stereology to count the number of DA neurons with TH staining in the injected part of the SNpc (Figure 1C,D). Observation at low-magnification showed no major reduction in the number of TH-positive neurons in any of the groups injected with AAVs encoding the LRRK2 fragments (Figure 1C). The total number of TH-positive cells in the SNpc did not differ significantly between the control group (PBS) and groups injected with AAV-ΔLRRK2^WT^, AAV-ΔLRRK2^G2019S^, or AAV encoding the dead kinase form ∆LRRK2^G2019S/D1994A^ (hereafter called ΔLRRK2^DK^) (Figure 1D). Thus, these results suggest that the ΔLRRK2 fragments alone did not trigger significant neurodegeneration of DA neurons at 15 weeks PI. The presence of motor asymmetry in the AAV-ΔLRRK2^DK^ group in the absence of neurodegeneration suggests that the sustained overexpression of this form can lead to cell dysfunction.

We wanted to investigate whether AAVs encoding the various ΔLRRK2 constructs could increase the toxicity of AA-α-syn^WT^ or AAV-α-syn^A53T^. We thus used an injection protocol that would lead to mild degeneration, such that a potential “pro-toxic” effect of the LRRK2 constructs could be easily detected. We conducted pilot experiments based on the literature to determine the appropriate dose (titers) of AAV-α-syn^WT^ and AAV-α-syn^A53T^ alone that would lead to progressive and partial degeneration of DA neurons in accordance with the time point determined in the previous experiment with ∆LRRK2 alone (Figure 1). The behavioral evaluation of the rats injected with AAV-α-syn^WT^ showed no significant motor asymmetry, and the quantification of TH-positive cells in the SNpc showed no significant reduction in the number of DA neurons with AAV-α-syn^WT^ at 15 weeks (not shown). However, AAV-α-syn^A53T^ led to statistically significant motor asymmetry in the cylinder test (Figure 2A) and a significant decrease in the number of TH-positive neurons (~45%) at 12 and 15 weeks after transduction with AAV-α-syn^A53T^ (2.5 × 10^10^ Vg) (Figure 2B,C). After injection of AAV-α-syn^A53T^, DA neurons often showed accumulation of α-syn phosphorylated at serine 129 (p-synS129). The cells that were positive for p-synS19 were also positive for ThioS, suggesting that a part of p-synS129 labeling corresponds to aggregates (Figure 2D).

Thus, a co-injection protocol with AAV-α-syn^A53T^ and AAV-∆LRRK2^G2019S^ and the evaluation of the number of TH-positive neurons at 15 weeks PI appeared to be suitable for the detection of the potential synergy of toxicity between the two pathological transgenes.

### 2.2. Effects of Co-Transduction with AAV-α-Syn^A53T^ and AAV-∆LRRK2^G2019S^

We next investigated whether the presence of the ΔLRRK2 fragments (G2019S or G2019S/D1994A-DK) in DA neurons could modify the toxicity of human α-syn^A53T^ using the co-transduction paradigm we developed (2.5 × 10^10^ Vg for each vector).

We first studied the neurotoxic effects produced by AAV-α-syn^A53T^ in the presence or absence of AAV-∆LRRK2^G2019S^ at 15 weeks PI. We assessed the co-localization of human α-syn^A53T^ and LRRK2 fragments in the SNpc after co-transduction, as we wanted to investigate the combined effects of α-syn^A53T^ and the various LRRK2 fragments in DA neurons. Analysis by confocal microscopy showed that the expression of human α-syn in the SNpc was high in TH-positive neurons (Figure 3A) and neurons expressing ΔLRRK2, as detected by the HA tag (Figure 3B). On average, 70% of neurons co-expressed both human α-syn^A53T^ and the LRRK2 fragments (Figure 3B, right panel).

The results of the cylinder test administered a few days before histological evaluation showed significant motor asymmetry in rats injected with AAV-α-syn^A53T^ alone or in combination with GFP (Figure 4A), whereas the rats that received AAV-α-syn^A53T^ combined with AAV-ΔLRRK2^G2019S^ showed no forepaw asymmetry (Figure 4A). Intriguingly, administration of methamphetamine did not produce asymmetrical rotation but rather an increase in the locomotor activity of the animals in all groups, which was significantly higher in the animals injected with AAV-α-syn^A53T^ combined with AAV-ΔLRRK2^G2019S^ than those injected with AAV-α-syn^A53T^ alone or combined with AAV-GFP (Figure 4B).

We next evaluated the toxic effects of human α-syn^A53T^ in the presence or absence of AAV-ΔLRRK2^G2019S^ (Figure 5A–F). AAV-α-syn^A53T^ alone produced a significant 38% decrease in the number of TH-positive cells, as measured by unbiased stereology in the SNpc at 15 weeks PI (mean count ± SEM: Control, 12,344 ± 734; AAV-α-syn^A53T^, 7555 ± 527) (Figure 5B).

The co-injection of AAV-α-syn^A53T^ with AAV-GFP (as a control of viral load) induced a 46% reduction in the number of DA neurons, which was not statistically different from that obtained with AAV-α-syn^A53T^ alone (6601 ± 360) (Figure 5B). The co-injection of AAV-α-syn^A53T^ and AAV-ΔLRRK2^G2019S^ induced a loss (−55%) of detectable TH-positive neurons (mean count ± SEM: 5585 ± 355), which was significantly greater than that measured in the two other groups injected with AAV-α-syn^A53T^ (Figure 5B).

We also counted the number of SNpc cells showing p-synS129 immunoreactivity, a marker of α-syn aggregation, in the various groups (Figure 5C,D). The number of p-synS129-positive cells was significantly lower in the group co-infected with AAV-α-syn^A53T^ and AAV-ΔLRRK2^G2019S^ than that in the groups infected with AAV-α-syn^A53T^ alone or in combination with AAV-GFP (Figure 5D). We also evaluated p-synS129 immunoreactivity in the striatum, which receives major inputs from the SNpc. Small p-synS129 immuno-positive objects with an elongated form or with a pearl necklace-like shape, reminiscent of neurite-like aggregates, were seen in the striatum (Figure 5E). Consistent with the results obtained in the SNpc, we found significantly lower levels of p-synS129 in the striatum of rats co-infected with AAV-α-syn^A53T^ and AAV-ΔLRRK2^G2019S^ than in those infected with AAV-α-syn^A53T^/GFP (Figure 5E,F).

Then, we evaluated the impact of SNpc degeneration on the level of DA terminals in the dorso-medial striatum using TH-immunofluorescence in both the α-syn^A53T^/GFP and α-syn^A53T^/∆LRRK2^G2019S^ groups at 15 weeks PI. These measurements were performed in the dorsal striatum (Figure 6A). TH immunoreactivity in the striatum in both α-syn^A53T^/GFP and α-syn^A53T^/∆LRRK2^G2019S^ groups was 15% lower than in the control group (PBS). This small α-syn^A53T^-induced loss of TH-positive fibers was similar in the GFP and ∆LRRK2^G2019S^ groups (Figure 6B,C).

### 2.3. Differential Effects of AAV-∆LRRK2^G2019S^ and AAV-∆LRRK2^DK^ on AAV-α-Syn^A53T^ Toxicity

We next investigated whether the effect of AAV-ΔLRRK2^G2019S^ on AAV-α-syn^A53T^-induced toxicity was dependent on the integrity of the kinase domain of the LRRK2 construct. We thus compared the effect of ΔLRRK2^G2019S^ with that of the dead kinase form AAV-∆LRRK2^DK^. We examined an early timepoint PI (6 weeks) for these experiments, before the appearance of motor alterations. We first carried out a semi-quantitative evaluation of the apparent levels of transgene expression six weeks after the injection of AAV-α-syn^A53T^ combined with AAV-ΔLRRK2^G2019S^ or AAV-∆LRRK2^DK^. Quantitative immunofluorescence analysis showed that almost the entire SNpc was infected by AAV-α-syn^A53T^ when co-infected with either AAV-ΔLRRK2^G2019S^ or AAV-∆LRRK2^DK^ (Figure 7A,B).

We also re-evaluated the co-localization of the ΔLRRK2-related transgenes and α-syn^A53T^ (Figure 8A,B). In total, 76% of neurons expressed both α-syn^A53T^ and ΔLRRK2 (Figure 8C, upper histogram), consistent with our observations in the experiments described above (see Figure 3). The relative expression levels of α-syn^A53T^ protein were the same in SNpc neurons co-expressing either ΔLRRK2^G2019S^ or ∆LRRK2^DK^ (Figure 8C, middle histogram). In addition, the relative expression of human ΔLRRK2^G2019S^ and that of ∆LRRK2^DK^ were similar in neurons (Figure 8C, bottom histogram).

The three groups showed no motor asymmetry in the cylinder test at this early PI timepoint (Figure 4). We then assessed the number of TH-positive neurons after infection with AAV-α-syn^A53T^ when co-injected with either AAV-ΔLRRK2^G2019S^ or ∆LRRK2^DK^. The reduction in the number of TH-positive neurons induced by human α-syn^A53T^ was significantly lower in the presence of ∆LRRK2^DK^ than in the presence of ΔLRRK2^G2019S^ (Figure 5G,H). The number of cells with p-synS129 immunoreactivity was similar in the ∆LRRK2^DK^ and ΔLRRK2^G2019S^ groups (Figure 5I,J). In the striatum, a few p-synS129 immunoreactive fibers (arrow heads), reminiscent of DA fibers, were seen in both groups that expressed α-syn^A53T^, with no apparent difference in density or size (Figure 5K,L).

We also carried out a preliminary characterization of the status of neuroinflammation at this early timepoint (6 weeks PI) by immunohistochemistry of Iba1, of which the expression is high in activated microglial cells. Indeed, there is a known role of neuroinflammation in α-syn^A53T^ rodent models [45]. As expected, microglial cells in rats overexpressing human α-syn^A53T^ appeared to be more reactive than those of rats injected with PBS (Figure 9A,C) or AAV-GFP (Figure 9E). The quantification of immunofluorescence in the SN (Figure 9B,F) and striatum (Figure 9D) showed that human α-syn^A53T^ significantly activated the microglia. However, overexpression of ΔLRRK2^G2019S^ or ΔLRRK2^DK^ did not have a major impact on the microglial activation induced by mutant human α-syn.

## 3. Discussion

The mechanisms leading to the degeneration of DA neurons in LRRK2 mutation gene carriers with PD are unknown. It is generally accepted that the LRRK2^G2019S^ mutation leads to increased kinase activity, which could then lead to cell death [24,26,46,47]. A role for α-syn in mutant LRRK2 toxicity has been suggested [33,34,35]. However, two important questions are still debated: (1) the mechanisms that underlie the interplay between LRRK2 and α-syn, and (2) if there is a functional interaction, the cell-autonomous and non-cell-autonomous mechanisms that are involved.

Here, we used an AAV-based approach to target SNpc DA neurons. AAV injections were performed unilaterally in the left SNpc. We investigated how the C-terminal domain of LRRK2, harboring the G2019S mutation, modifies the toxic effects induced by the overexpression of α-syn^A53T^ toward DA neurons in the rat SNpc. Under these experimental conditions, AAV-ΔLRRK2^G2019S^ alone did not induce a decrease in DA neuron numbers or major motor asymmetry at the time points PI that we studied. Intriguingly, we found that -ΔLRRK2^DK^, while producing no apparent loss of TH-positive cells, could induce motor asymmetry, suggesting it may produce cellular disturbances. There are only a few studies providing behavioral characterizations of transgenic rodent models expressing the dead kinase form of LRRK2^G2019S^, but none reported a major behavioral abnormality. In contrast, AAV-α-syn^A53T^ alone induced a significant decrease in the number of TH-positive neurons and a significant motor asymmetry of the forepaws as assessed by the cylinder test, one of the most sensitive tests used to detect motor deficits in unilateral animal models of PD [48,49]. Intriguingly, we observed that, although co-expression of ΔLRRK2^G2019S^ and α-syn^A53T^ induced the significant reduction of DA neuron number at 15 weeks PI, there was no forepaw asymmetry. In the animals injected with AA-α-syn^A53T^ alone (or with AAV-GFP), we observed significant motor asymmetry, whereas the reduction in the number of DA neurons was smaller than that seen in rats injected with ΔLRRK2^G2019S^ and α-syn^A53T^. As these findings were counter-intuitive, we also tested the effect of an injection of methamphetamine in the various groups to indirectly study the functional “state” of the nigro-striatal DA pathway. Amphetamine administration did not induce rotational behavior in these animals. This is consistent with the limited reduction in DA neuron number in the SNpc and very limited decrease in DA fibers in the striatum, in contrast to what occurs in models of profound DA cell loss (e.g., 6-OHDA). However, amphetamine administration induced an increase in locomotor activity in all groups, as seen by an increase in the number of turns during the 90 min trial. Interestingly, locomotion was higher in rats injected with AAV-ΔLRRK2^G2019S^ and AAV-α-syn^A53T^ than those injected with AAV-α-syn^A53T^ alone or in combination with AAV-GFP. This suggests a higher release of DA in rats injected with AAV-ΔLRRK2^G2019S^ and AAV-α-syn^A53T^ than for the other groups. This interpretation awaits further experiments aiming at determining extracellular DA concentrations in the striatum using microdialysis or cyclic voltammetry in rats infected with the different AAVs. This is reminiscent of the increase in DA turnover detected using PET scans in presymptomatic subjects with a mutant LRRK2 gene [50]. Histological evaluation of our rat models showed that the combination of AAV-ΔLRRK2^G2019S^ plus AAV-α-syn^A53T^ induced a significant but extremely small decrease in TH immunoreactivity in the striatum in the animals injected with AAV-α-syn^A53T^ and AAV-GFP. Thus, the difference in the sensitivity to amphetamine administration correlated with neither the level of TH expression in the striatum nor the severity of DA neuron degeneration in the SNpc. Indeed, the reduction in the number of neurons in the group injected with AAV-ΔLRRK2^G2019S^ plus AAV-α-syn^A53T^ was higher than in the group injected with AAV-α-syn^A53T^ alone or along with AAV-GPF. It is likely that neurodegeneration occurred earlier in the group injected with the two vectors than the group injected with AAV-α-syn^A53T^ alone or along with AAV-GPF. Indeed, the injection of AAV-ΔLRRK2^G2019S^ plus AAV-α-syn^A53T^ appeared to already be more toxic to DA neurons than that of AAV-α-syn^A53T^ alone (or with AAV-GFP) at 6 weeks PI. The lack of correlation between neurodegeneration and motor alterations may result from the instauration of compensatory mechanisms in the DA neurons that survived and/or which are not transduced.

The present experimental paradigm using AAVs allowed us to address the question of the potential cell-autonomous exacerbation of α-syn^A53T^ toxicity by the kinase activity of LRRK2 directly in the SNpc and only in neurons. In contrast, other viral vector platforms that could potentially host the full-length LRRK2 ORF (i.e., vectors derived from herpes simplex virus (HSV) or adenovirus) also transduce other cell types in the striatum [51,52]. Here, we directly investigated whether there is a functional interaction between AAV-α-syn^A53T^ and AAV-ΔLRRK2^G2019S^ in DA neurons. Our results show the existence of such a “functional” interaction, as overexpression of ΔLRRK2^G2019S^ significantly enhanced the neurotoxic effects of α-syn^A53T^ in the rat SNpc. Lin et al. showed that the overexpression of LRRK2 (WT or with the G2019S mutation) in forebrain neurons (striatum and cerebral cortex) increased the toxicity of α-syn^A53T^ in transgenic animals [35]. In these double-transgenic mice, the authors found significant degeneration of the striatum and cortex and enhanced accumulation of α-syn aggregates. This proved the existence of functional crosstalk between α-syn and LRRK2 in neurons in vivo when the proteins are expressed at relatively high levels. Pathological transgenes were not expressed in the SNpc, and DA degeneration was not observed in these models. The CamKIIα promoter used to drive the expression of the tetracycline transactivator (tTA), which activates the TetO promoter of the LRRK2 and α-syn^A53T^ transgenes in these mice, is likely not active in SNpc DA neurons, as endogenous expression of CamKIIα in neurons of the SNpc is lower than that observed in forebrain neurons ([53] and see also the Allen Brain Atlas, http://mouse.brain-map.org/experiment/show/79490122 accessed on 10 June 2021). In LRRK2 knockout rats, the toxicity induced by AAV encoding α-syn is lower than in WT rats [54]. Daher et al. found no synergy between the transgenes following the crossbreeding of other transgenic models in which the promoters driving LRRK2^G2019S^ and α-syn^A53T^ expression were different (Prion and CMV, respectively) [54]. Indeed, data from the latter study indicate that the expression of the human LRRK2 transgene is low in the SNpc (see Figure 2 in [54]). Neurons that express α-syn^A53T^ are apparently sparse in the SNpc relative to the known density of DA neurons in this structure (see Figure 5 in [54]). In a more recent study, the question of whether LRRK2^G2019S^ toxicity occurs in a kinase-dependent manner in DA neurons was addressed using tetracycline (Tet)-inducible conditional transgenic (Tet-LRRK2^GS^) and kinase-dead (GS/DA) mice under the control of a human TH promoter [55]. These models reveal an age- and kinase-dependent neurodegeneration of DA and norepinephrine neurons accompanied with the accumulation of pathological, endogenous α-syn, supporting the hypothesis that mutant LRRK2 contributes to α-syn pathology. These various studies and our results suggest that the crosstalk between LRRK2 and α-syn can only occur if the two proteins are localized to the same neurons.

The interplay between ΔLRRK2^G2019S^ and α-syn^A53T^ was detected under conditions in which the two proteins were expressed at high levels. We could not precisely compare the levels of the proteins expressed from the transgenes, which are of human origin, to those of the endogenous rat proteins. It can be only grossly estimated that the expression of human ΔLRRK2^G2019S^ and α-syn^A53T^ was likely 10- to 50-fold higher than that of the endogenous rat proteins. This estimation is based on previous experiments in which other mouse transgenes (*Dclk3*, *Crym*, *abhd11os*, *Capucin*) were overexpressed with lentiviral vectors or AAVs [56,57,58,59,60]. Thus, we cannot rule out that the cell-autonomous crosstalk between LRRK2 and α-syn may be of only moderate importance in DA neurons when the two proteins are expressed at physiological levels.

The relevance of overexpressing the C-terminal domain of LRRK2 versus the full-length protein is debatable, and the mechanisms underlying the neurotoxic effect of ΔLRRK2^G2019S^ in our models are unknown. Indeed, our ΔLRRK2^G2019S^ construct lacked the N-terminal domains, which are known to play crucial roles in LRRK2 function. We previously showed that overexpression of the ΔLRRK2^G2019S^ fragment using AAVs triggers neurodegeneration of DA neurons six months PI, whereas the ΔLRRK2^WT^ fragment, expressed at similarly high levels, was devoid of obvious neurotoxicity [31]. In the aforementioned study, we suggested that the death of DA neurons induced by ΔLRRK2^G2019S^ is likely independent of any interaction with RAB10, as we found that the ΔLRRK2 fragment was unable to interact with RAB10, in contrast to the full-length LRRK2 fragment [31]. Thus, other signaling pathways have to be considered. It is conceivable that the overexpression of ΔLRRK2^G2019S^ leads to abnormally high phosphorylation of substrates relative to ΔLRRK2^WT^. Indeed ΔLRRK2^G2019S^ kinase activity is higher than that of ΔLRRK2^WT^ [32], a phenomenon that is also observed for full-length LRRK2^G2019S^ [61,62,63,64]. Alternatively, the “pro-toxic” effect of ΔLRRK2^G2019S^ on α-syn^A53T^ could also result from molecular mechanisms unrelated to the enzymatic activity of the catalytic domains. Changes in protein–protein interactions and/or a modification of the conformation of LRRK2 fragments induced by the G2019S substitution may also play a role. In support of this hypothesis, LRRK2 interacts with microtubules [65,66], and recent high-resolution cryo-EM studies have shown that the enzymatic domain of LRRK2 (ROC-COR-Kinase) is sufficient for the interaction of LRRK2 with microtubules and their regulation [67,68]. The orientation of the kinase domain relative to microtubules is different between WT LRRK2 and LRRK2 with pathological mutations [68]. Therefore, it is conceivable that the pro-toxic effect of ΔLRRK2^G2019S^ in our experiments is linked to microtubule-related perturbations. Further in vivo studies are required to fully address this hypothesis.

LRRK2 fragments are expressed only in neurons in our AAV-based model, which allowed us to investigate the cell-autonomous mechanisms of LRRK2/α-syn interplay. The other comparable experimental approaches that investigated this interplay were carried out in models in which LRRK2 is expressed in all cells (see Daher et al. [34]). In transgenic animal models and patients, LRRK2 is expressed in cells of various types (i.e., neurons, microglia, oligodendrocytes, and astrocytes). Under such conditions, non-cell-autonomous mechanisms involving the interaction of DA neurons with neighboring glial cells and immune cells may likely have important roles. For example, it has been recently shown that the seeding of α-syn aggregates by the exposure of neurons to α-syn fibrils is higher in neurons that express mutant LRRK2 [69]. More generally, LRRK2 mutation may change the potential propagation of aggregated α-syn species in the brain [70]. The level of LRRK2 activity in microglial cells may also regulate pro-toxic phenomena associated with α-syn-induced neuroinflammation [31,41]. LRRK2 plays a key role in the immune system [71]. A single nucleotide polymorphism (N2081D) in the region encoding the kinase domain of LRRK2 is a major risk factor for Crohn’s disease, a form of inflammatory bowel disease [72]. Indeed, although α-syn^A53T^ was primarily expressed in neurons with the AAV and promoter we used in our study, we expected to observe increased activation of microglial cells if ΔLRRK2^G2019S^ increases the propagation of α-syn^A53T^ from neurons to other cells. Our quantitative characterization of Iba1-positive cells induced by AAV injection in the SNpc indeed showed microglial activation linked to human α-syn^A53T^. Notably, the AAV vehicle and AAV-GFP did not produce such microglial activation. Microglial activation was not altered by the ΔLRRK2 fragments.

Our results also indicate that ΔLRRK2^G2019S^ does not markedly change p-synS129 immunoreactivity at 6 weeks PI. The reduction of the number of p-synS129-positive cells in the SNpc in the group expressing AAV-ΔLRRK2^G2019S^ and AAV-α-syn^A53T^ PI is possibly linked, at least in part, to the reduction in the number of DA neurons at 15 weeks. Indeed, the apparent loss of SNpc neurons positive for p-synS129 is proportional to the degeneration of TH-positive neurons in the SNpc in groups injected with PBS, AAV-GFP, and AAV-ΔLRRK2^G2019S^.

The apparent incoherence in the results relative to the small reduction (~15%) in the apparent density of striatal TH-positive fibers and the more profound reduction (~60%) of striatal p-syn129-positive objects has no definitive explanation. The disappearance of striatal TH-positive fibers and the presence of p-synS129 are produced by the expression of α-syn^A53T^. One might expect that the reduction of TH-positive fibers at 15 weeks PI should lead to a proportional reduction in the accumulation of striatal p-syn129-positive objects. However, we observed that the reduction in the accumulation of p-syn129-positive objects is more profound than the apparent loss in TH-positive fibers in the striatum in rats co-infected with AAV-ΔLRRK2^G2019S^ and AAV-α-synA53T as compared to those infected with AAV-α-synA53T only. This might reflect other mechanisms. For example, it is conceivable that ΔLRRK2^G2019S^ produces defects in axonal transport, reducing the anterograde transport of α-syn aggregates from the neuronal bodies to axons. Indeed, LRRK2^G2019^ can affect microtubules and axonal transport [65]. These results suggest that the “pro-toxicity” induced by the overexpression of ΔLRRK2^G2019S^ may not be directly related to changes in the bioavailability, elimination, synthesis, or aggregation rates, although more complete biochemical studies are required to further explore this question.

Thus, these novel findings are consistent with the hypothesis that the C-terminal domain of LRRK2^G2019S^ is sufficient to augment the neurotoxic effects of α-syn^A53T^ through a cell-autonomous mechanism involving the kinase domain, not directly linked to a major modification of α-syn aggregation and/or exacerbation of the α-syn^A53T^–induced microglial response.

Only a few studies have directly addressed the role of the kinase domain in the interaction between α-syn and LRRK2 toxicity. It is generally accepted that the LRRK2^G2019S^ mutation leads to increased kinase activity, which could then lead to cell death [46,47,57]. Here, we found that the inactive protein ΔLRRK2^DK^ did not markedly alter the toxicity of AAV-α-syn^A53T^ at 6 weeks PI, whereas ΔLRRK2^G2019S^ increased the toxicity of AAV-α-syn^A53T^ towards DA neurons. Pioneering studies showed that the neuroinflammation and neurodegeneration produced by transduction of the SNpc with AAV-α-syn are significantly attenuated in LRRK2 KO rats relative to those in WT littermates. In these experiments, the role of the kinase activity was not assessed [33]. More recently, Daher et al. showed the toxicity of AAV-α-synuclein in the SNpc to be higher in transgenic LRRK2^G2019S^ than WT rats. These results can be explained by (1) the higher kinase activity of LRRK2^G2019S^ or, alternatively, (2) a subtle structural change in the C-terminal domain that modifies the function of the entire protein toward itself (e.g., autophosphorylation) or protein partners. In support of the first hypothesis, pharmacological inhibition of LRRK2 suggests a role of the catalytic activity of the kinase domain in the toxicity of the protein. Indeed, treatment of LRRK2^G2019S^ rats with the LRRK2 inhibitor PF-06447475 reduces the toxicity of α-syn [34]. However, it has been observed that certain LRRK2 inhibitors, including PF, can reduce cellular levels of the protein [73]. In vitro cell-culture experiments have shown that the level of expression of the LRRK2 protein may play a determinant role in mutant LRRK2 toxicity rather than its kinase activity [36]. Thus, it is possible that protection by PF-06447475 against the toxicity triggered by injection of AAV-α-syn in LRRK2^G2019S^ animals may result from a reduction in LRRK2 levels rather than actual inhibition of the catalytic activity of the kinase. However, new-generation inhibitors with protective effects do not reduce LRRK2 levels (see for review [37]). We show here that the pro-toxic effect of ΔLRRK2^G2019S^ (not observed for ΔLRRK2^DK^) in neurons expressing α-syn^A53T^ is not related to a major difference in the level of expression of the protein in vivo. Our semi-quantitative confocal analysis showed no major change in the α-syn^A53T^ relative levels between the ΔLRRK2^G2019S^ and ΔLRRK2^DK^ groups. Additional ultra-high-resolution microscopy experiments would be necessary to rule out subtle subcellular changes in expression or redistribution of α-syn^A53T^. Our results do not allow us to state with certainty whether the catalytic activity of ΔLRRK^2G2019S^ is central to its effect on α-syn^A53T^ or whether other molecular mechanisms are involved.

Our results show that the C-terminal domain of LRRK2^G2019S^ containing the ROC-COR, kinase, and WD40 domains is sufficient to potentiate the toxicity of human α-syn^A53T^ in DA neurons in vivo and suggest that this effect depends on the kinase domain. This cell-autonomous mechanism may act additively or synergistically along with other non-cell-autonomous mechanisms.

## 4. Materials and Methods

### 4.1. Adeno-Associated Viral Vectors (AAVs) Construction and Production

AAV6 viral particles were obtained by encapsidation of AAV2 recombinant genomes into serotype 6 AAV capsids, as described previously [74]. Briefly, viral particles were produced by co-transfection of HEK-293T cells with (1) an adenovirus helper plasmid (pXX6-80), (2) an AAV packaging plasmid carrying the rep2 and cap6 genes, and (3) a plasmid encoding a recombinant AAV2 genome containing the transgene expression cassette. Seventy-two hours following transfection, viral particles were purified and concentrated from cell lysates and supernatants by ultracentrifugation on an iodixaniol density gradient, followed by dialysis against PBSMK (0.5 mM MgCl2 and 1.25 mM KCl in PBS). The concentration of vector stocks was estimated by real-time PCR following the method described by Aurnhammer et al. [75] and expressed as viral genomes per ml of concentrated stocks (Vg/ml). AAVs encoding human ∆LRRK2 (WT, G2019S, and G2019S plus D1994A mutation, i.e., “kinase-dead”), α-syn^A53T^, and GFP under the PGK1 (mouse phosphoglycerate kinase) promoter were produced.

### 4.2. Stereotaxic Injection

Adult Sprague–Dawley rats (Charles River Laboratories), weighing ~250 g (Charles River, Saint Germain sur l’Arbresle, France), were housed under a 12 h light/dark cycle with ad libitum access to food and water, in accordance with European Community (Directive 2010-63/EEC) and French (Code Rural R214/87-130) regulations. Experimental procedures were approved by the local ethics committee and registered with the French Research Ministry (committee #44, approval #12-100, and APAFIS#1372-2015080415269690v2). For stereotaxic injections, the animals were deeply anaesthetized with 4% isoflurane, followed by a mixture of ketamine (75 mg/kg) and xylazine (5 mg/kg), and placed in a stereotaxic frame. Recombinant AAVs were injected unilaterally into the SNpc at the following stereotaxic coordinates: +3.4 mm anterior to the interaural zero and ±2.0 mm lateral to the bregma, at a depth of −7.8 mm relative to the skull, with the tooth bar set at −3.3 mm. We injected 4 µL of virus at a concentration of 2.5 × 10^10^ Vg per site for single injections and 2.5 × 10^10^ Vg of each vector for co-injections for a total of 5 × 10^10^ Vg per site, with a 34-gauge blunt-tipped needle linked to a 10 μL Hamilton syringe by a polyethylene catheter at a rate of 0.25 μL/min using an automatic pump (CMA-4004). The needle was left in place for five minutes and then slowly withdrawn.

### 4.3. Evaluation of Motor Behavior

In this study, we combined the two most common and sensitive behavioral tests used to assess motor deficiency as is recommended by Anders Bjorklund and Stephen B. Dunnett [49] and Simon P. Brooks and Stephen B. Dunnett [48]. The cylinder test was used to assess dissymmetry induced by unilateral injection of the AAV vectors. Rats were placed in a transparent cylinder for 5 min and recordings performed for the duration of the test using a camera placed below the cylinder. The animals generally straighten out and explore their environment by touching the side walls of the cylinder. The recorded films were then viewed at a lower speed, and the number of times the animal touched the walls with only the left, right, or both legs simultaneously was counted. The data are reported as the percentage of use of the contralateral paw using the following formula: [(contralateral + ½ both)/(ipsilateral + contralateral + both)] × 100, previously used by Gombash et al. [44].

Amphetamine is a dopamine-stimulating drug. If there is unilateral injury of the SN*pc* or striatum, an intraperitoneal injection of amphetamine induces anti-clockwise rotation behavior if the lesion is performed in the left hemisphere. Rats received an intraperitoneal injection of (+)-Methamphetamine hydrochloride C-IIN (2.5 mg/kg, M8750, Sigma) and were then hooked to harnesses and placed in a cylinder. This system automatically records the number of revolutions that the animal performs over 90 min (diameter: 40 cm, Bioseb, Multicounter LE806).

### 4.4. Tissue Processing

For all procedures, rats were first deeply anesthetized by isoflurane inhalation, followed by intraperitoneal injection of a lethal dose of sodium pentobarbital. Rats were transcardially perfused with 300 mL 4% paraformaldehyde (4% PFA) in phosphate buffer saline (PBS-0.1 M phosphate buffer, 9 g/L NaCl) at a rate of 30 mL/min. After perfusion, the brain of each rat was quickly removed and immersed in ice-cold 4% PFA/PBS for at least 24 h, before transfer to 15% sucrose in PBS for 24 h and then 30% sucrose in PBS the next day, for cryoprotection. The brains were then cut into 40-μm sections on a freezing microtome (SM2400, Leica, Wetzlar, Germany). Serial sections of the striatum and midbrain were stored in antifreeze solution (30% glycerol/30% ethylene glycol in PBS) and stored at −20 °C until use.

### 4.5. Immunohistological Analysis and Quantification

#### 4.5.1. Immunohistochemistry

Sections were removed from the antifreeze solution and washed in PBS. Endogenous peroxidase activity was quenched by transferring them to 1% H_2_O_2_, incubation for 30 min at room temperature (RT), and washing three times with PBS for 10 min. The sections were then blocked by incubation with 4.5% normal goat serum for 30 min in PBS-T (0.2% Triton X-100 in PBS) and then incubated overnight with primary antibody in 3% normal goat serum in PBS-T at 4 °C with gentle shaking.

For histological evaluation of rat brain sections, the following primary antibodies were used: anti-tyrosine hydroxylase (TH) antibody, MAB318 clone LNC1, Merk-Millipore, 1:3000; anti-hemagglutinin tag (HA), Covance clone 11, 1:1000; anti-human α-synuclein, syn 211, 1:1000; and anti-phospho-α-synS129, ab51253, Abcam, 1:5000. The next day, the sections were removed from the primary antibody solution, washed three times, and incubated for 1 h at RT with the appropriate biotinylated secondary antibody in PBS-T (Vector Laboratories, Burlingame, CA, USA, 1:1000). The sections were then washed and incubated with ABC complex solution in PBS-T (1:250, reagents A and B combined in a 1:1 ratio, Vector Laboratories) for 1 h.

The rat brain sections were then incubated with diaminobenzidine for 30 s to 1 min and, after dehydration, mounted on slides in Eukitt mounting medium.

#### 4.5.2. Cell Counting

Optical fractionator sampling was carried out on a Zeiss AxioPlan microscope. Midbrain DA neurons were outlined on the basis of TH immunolabeling with reference to a coronal atlas of the rat brain (Paxinos and Watson, 6th edition). TH-positive cells were counted by unbiased stereology in the entire SNpc and the number of positive neurons per section was calculated using Mercator Software (Explora Nova, France). We placed 100 × 100 μm grids in a systematically random manner, 80 × 80 μm apart, with a 3 µm offset from the surface of the section. Quantification was performed on 12 serial sections spaced by 200 µm, corresponding to the entire SNpc.

The phosphorylation of α-syn on S129 (p-synS129) was evaluated by counting the number of p-synS129-positive neurons in the SNpc using stereology methods. The SNpc was delimited by Nissl staining and the grids (250 × 250 µm) placed with a spacing of 100 × 100 µm. Quantification was performed on six serial sections spaced by 400 µm, corresponding to the entire SNpc. In the striatum, a threshold was applied to select only the p-synS129-positive neurons by immunostaining and quantification performed on three slices, corresponding to the beginning, middle, and end of the striatum.

#### 4.5.3. Immunofluorescence

The procedure used was similar to that for immunohistochemistry, but without incubation in 1% H_2_O_2_. The primary antibodies used for the immunofluorescence procedure were the same as those already described (IBA1, Wako, 1:1000). Sections were first incubated with the primary antibody overnight at 4 °C. The next day, they were incubated with a fluorescent secondary antibody (Alexa Fluor 594-labeled goat anti-rabbit IgG or Alexa Fluor 488-labeled goat anti-rabbit IgG (1:1000, Life Technologies)) for 1 h at RT. Sections were then washed and incubated overnight at 4 °C with another primary antibody. Finally, they were incubated with a second fluorescent secondary antibody (Alexa Fluor 488-labeled goat anti-mouse IgG or 594-labeled goat anti-mouse IgG (1:1000, Life Technologies)) for 1 h at RT. The sections were stained with DAPI, washed, and mounted in a fluorescence mounting medium. Images were acquired with a laser confocal microscope (SP8, Leica, Wetzlar, Germany) or an epifluorescence microscope (DM6000, Leica, Wetzlar, Germany).

#### 4.5.4. Thioflavin-S Staining

A double-staining protocol was used to verify that the accumulated p-synS19 inside cells could colocalize with the aggregated form of α−syn. The immunostaining procedure for p-synS19 and DAPI staining was performed on floating sections before the thioflavin-S (Thio-S) staining. Floating sections were washed in PBS and mounted on Superfrost Plus slides. Slides were placed in holders and dipped into 70% and then 80% EtOH, for 1 min each. Then, the slides were incubated in Thio-S diluted to 1% in distilled water for 7 min. The Thio-S solution must be protected from light, filtered before use, and stored at 4 °C. Then, slides were washed in 80% EtOH, 70% EtOH, and distilled water for 1 min each before being cover-slipped with the fluorescence-mounting medium.

#### 4.5.5. Colocalization

The percentage of co-localization between ΔLRRK2 and α-syn was determined by counting the number of cells co-expressing both ΔLRRK2 and α-syn proteins divided by the number of cells expressing α-syn alone. Images were acquired with a laser confocal microscope (SP8, Leica, Wetzlar, Germany). The percentage of co-localization between ΔLRRK2 and α-syn was determined by counting the number of cells co-expressing both ΔLRRK2 and α-syn proteins divided by the number of cells expressing α-syn alone. Images were acquired with a laser confocal microscope (SP8, Leica, Wetzlar, Germany).

The relative levels of ΔLRRK2 and α-syn proteins were evaluated on three coronal sections in the SNpc. All sections were incubated in parallel with the same buffers containing primary (anti-hemagglutinin tag (HA), Covance clone 11, 1:1000; anti-human α-synuclein, syn 211, 1:1000) and secondary antibodies (Alexa Fluor 594-labeled goat anti-rabbit IgG or Alexa Fluor 488-labeled goat anti-rabbit IgG (1:1000, Life Technologies, Aurora, CO, USA) to reduce as much as possible inter-animal and inter-section (sample) variation in immunostaining. Cells expressing human α-synA53T and/or ΔLRRK2-HA were acquired, taking care not to change the tuning of the microscope between fields of view and animals. To compare the levels of expression of both human proteins within the same cells, 20 cells were analyzed for each animal. The evaluation of light intensity in the different channels reflects measures of 180 cells (5–4 rats randomly chosen per group). Images acquired as 0.5 µm Z-stacks were summed to produce an image of total light intensity over the Z-axis. Cells were delineated manually using image J software and the mean fluorescence intensity in the red and green channels (corresponding to ΔLRRK2 and α-syn proteins, respectively) was measured in each cell. To determine the number of cells that would be sufficient to evaluate the mean of cell intensities, we conducted a statistical analysis comparing the two groups. With a sampling of 20 cells per rat, the dispersion of measures of light intensity for the 20 cells in each animal is similar to the dispersion of mean measures in each group. More precisely, the standard deviations (STD) of the measures of light intensity determined in each animal for α-syn (~40% of the cell mean) and ΔLRRK2 (43% of the cell mean) immunofluorescence were in the range of the STD of the mean observed in the two groups (α-syn, ~36% of the mean; ΔLRRK2, ~22% of the mean).

#### 4.5.6. Epifluorescence Intensity Measurement

Striatal DA innervation at 15 weeks was evaluated by measuring the fluorescence intensity of TH-immunoreactive terminals on three coronal striatal sections. The sections were observed by epifluorescence microscopy at a magnification of 63× and the fluorescence intensity was determined using MorphoStrider software (Explora Nova, La Rochelle, France).

#### 4.5.7. Microglia Area Measurement

The area occupied by microglia was evaluated by confocal microscopy at a magnification of ×20 in the dorso-medial part of the striatum and in the SN pars reticulata. A threshold was applied and the area of 20 microglia cells measured per acquisition. Three acquisitions per animal were used.

#### 4.5.8. Statistical Analysis

The normality of data distribution was tested using the Shapiro–Wilk test and the homogeneity of variance with Levene’s test using commercially available software (Statistica, 13.0; Statsoft Inc., Tulsa, OK, USA). When the criteria of normality and homogeneity of variance were met, unpaired Student’s *t*-tests were used for pairwise comparisons between groups. One-way analysis of variance (ANOVA) for multiple comparisons was carried out for comparisons of more than two groups, with Fisher’s post hoc PLSD test. In cases in which the assumption of normality and/or homogeneity of variance were not met, non-parametric tests were applied: Mann–Whitney and Kruskall–Wallis for comparison of two or more groups, respectively. The annotations used to indicate the level of significance are as follows: * *p* < 0.05, ** *p* < 0.01, *** *p* < 0.001.

## Figures and Tables

**Figure 1 ijms-22-06760-f001:**
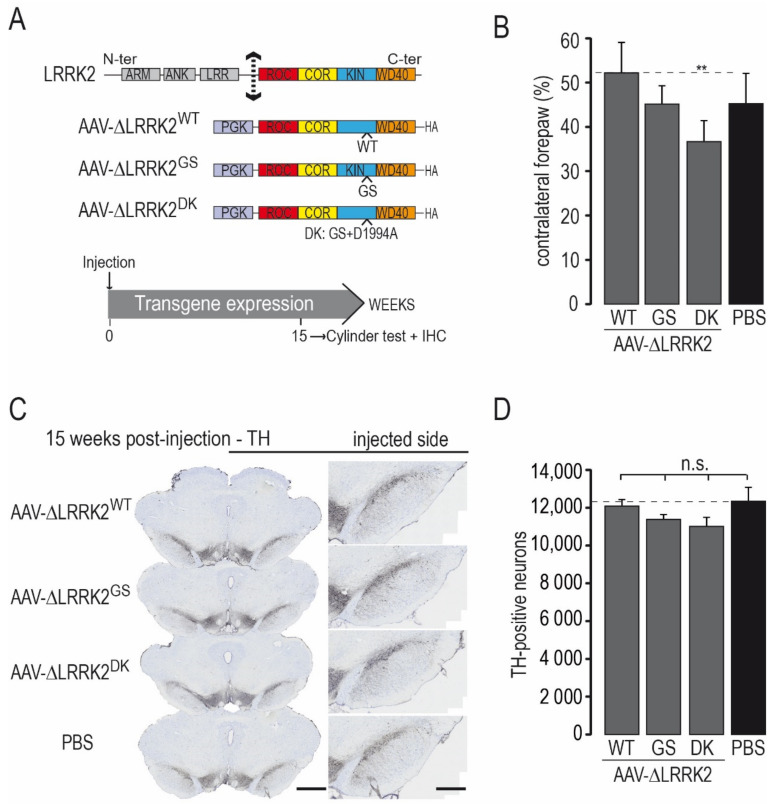
Effects produced by intra-nigral injection of AAVs coding for various forms of the C-terminal fragment of LRRK2. (**A**) Various forms of the C-terminal fragment of LRRK2 (∆LRRK2) were cloned into an AAV backbone with the PGK promoter: the wild-type form (WT), the pathological form with the G2019S S substitution (GS), or the dead kinase form of G2019 with the D1994A mutation (DK). AAVs were unilaterally injected into the rat SNpc. The cylinder test, which assesses forepaw asymmetry use, was performed at 15 weeks post-injection (PI), and the rats were processed for histological evaluation (ICH). (**B**) Results of the cylinder test at 15 weeks PI. (**C**) Representative photomicrographs of sections of the various experimental groups labeled using Tyrosine Hydroxylase (TH) immunohistochemistry. (**D**) Number of TH-positive neurons in the SNpc measured using unbiased stereology. Results are expressed as the means ± the SEM. N = 8–12 animals/group. ANOVA and PLSD post hoc test. n.s.: not significant. ** *p* < 0.01. Scale bars: 750 µm left panel and 400 µm right panel in (**C**).

**Figure 2 ijms-22-06760-f002:**
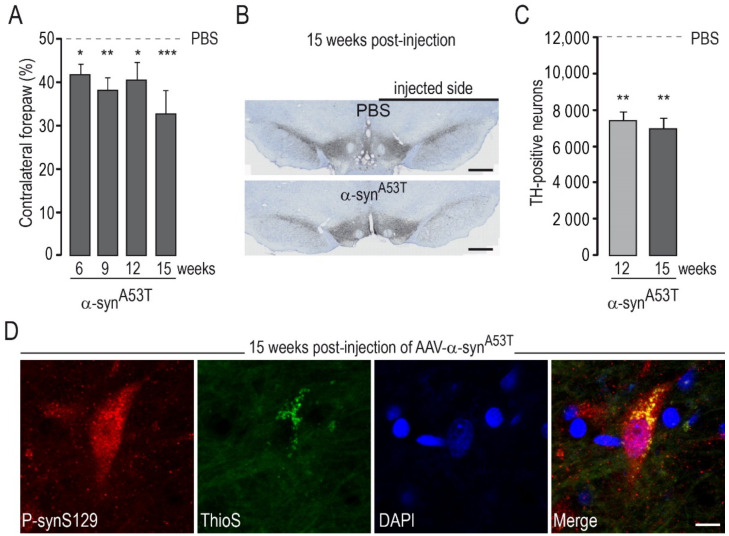
Degeneration and motor symptoms produced by intra-nigral injection of AAVs encoding α-syn^A53T^. AAV-α-syn^A53T^ (2.5 × 10^10^ Vg) or vehicle (PBS) were unilaterally injected into the rat SNpc. The cylinder test, which assesses asymmetry of forepaw use, was performed at various timepoints (6–15 weeks) post-injection (PI). Two subgroups of rats were processed for histological evaluation (ICH) at 12 and 15 weeks PI. (**A**) Results of the cylinder test at various time points after AAV injection. (**B**) Representative photomicrographs of the SNc in rats injected with AAV-α-syn^A53T^ or vehicle (PBS) labeled by Tyrosine Hydroxylase (TH) immunohistochemistry. (**C**) Number of TH-positive neurons in the SNpc measured using unbiased stereology showing a consistent decrease in the number of TH-positive neurons. (**D**) Representative confocal images obtained by double immunofluorescence analysis in the SNpc of rats injected with AAV-α-syn^A53T^ at 15 weeks PI: neurons with α-syn phosphorylated at serine 129 (p-synS129) (in red). The neuron with high levels p-synS19 immunoreactivity is also positive for ThioS (in green), suggesting that p-synS129 accumulation corresponds, at least partially, to aggregated forms of α-syn. Results are expressed as the means ± the SEM. N = 8–12 animals/group. ANOVA and PLSD post hoc test. * *p* < 0.05, ** *p* < 0.01, and *** *p* < 0.001. Scale bars: B, 750 µm; D, 10 µm.

**Figure 3 ijms-22-06760-f003:**
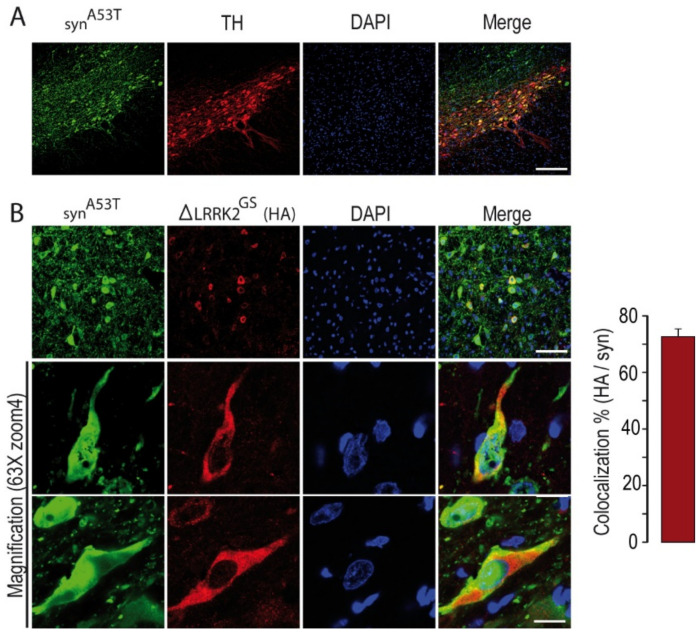
Histological evaluation of the expression of the transgenes in the SNpc at 15 weeks post-injection. (**A**) Evaluation of α-syn (in green) transduction in the SNpc after co-injection of AAV-α-syn^A53T^ with ΔLRRK2^G2019S^ (ΔLRRK2^GS^) as determined by delineation of SNpcs with TH staining (red). Scale bar: 500 µm. (**B**) Measurement of the number of neurons expressing both α-syn^A53T^ and ΔLRRK2^GS^ from confocal images. The higher magnification shows cytoplasm localization of ΔLRRK2^G2019S^.

**Figure 4 ijms-22-06760-f004:**
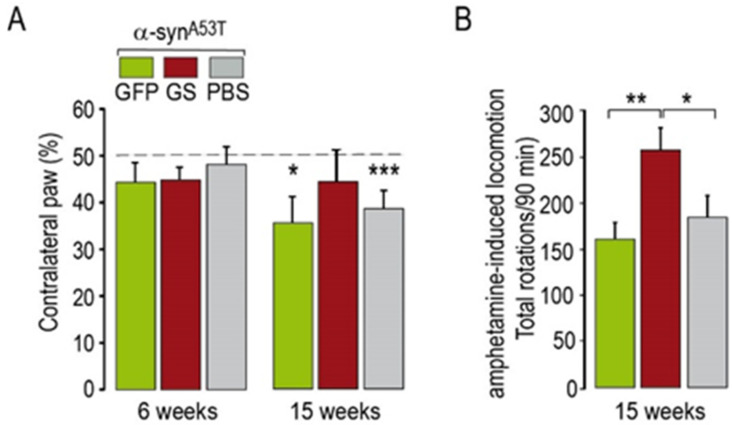
Motor tests in rats co-injected with AAV-α-syn^A53T^ and AAV-GFP or AAV-α-syn^A53T^ and AAV-ΔLRRK2^G2019S^ (ΔLRRK2^GS^). (**A**) Rats were tested using the cylinder test at 6 and 15 weeks PI to detect asymmetry in forepaw use. (**B**) Rats were injected with methamphetamine (2.5 mg/kg) to induce hyper-locomotion. Results are expressed as the means ± SEM. N = 8–12 animals/group. ANOVA and PLSD post hoc test. * *p* < 0.05, ** *p* < 0.01, *** *p* < 0.001.

**Figure 5 ijms-22-06760-f005:**
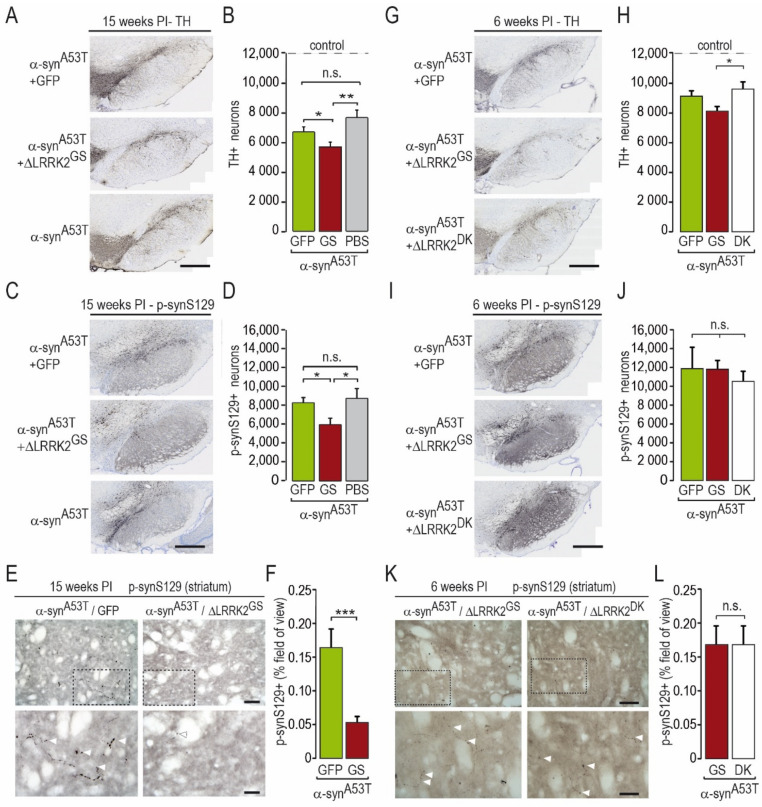
Immunohistochemistry for tyrosine hydroxylase (TH) and p-synS129-positive (p-synS129) cells and axons at 15 and 6 weeks post-injection. (**A**–**C**) Histological images and graphical quantification of TH-positive cells in the SNpc at 15 weeks PI (**A**,**B**) and 6 weeks PI (**G**,H). (**E**–**H**) Histological images and graphical quantification of p-synS129-positive (p-synS129+) neurons in the SNpc at 15 weeks PI (**C**,**D**) and 6 weeks PI (**I**,**J**). The number of TH-positive and p-synS129-positive cells was evaluated using unbiased stereology at very high magnification. (**I**–**L**) Histological images and graphical quantification of rat brain sections labeled by p-synS129 immunohistochemistry at the level of the striatum at 15 weeks PI (**E**,**F**) and 6 weeks PI (**K**,**L**) showing the presence of sparse positive objects with a necklace-like organization. The quantification was determined as the percentage of the field of view (area) occupied by p-synS129-positive staining. Results are expressed as the means ± SEM. N = 7–13 animals/group. ANOVA and PLSD post hoc test. * *p* < 0.05, ** *p* < 0.01, *** *p* < 0.001. Scale bars: 400 in (**A**,**C**,**E**,**G**) and 50 µm in images in (**I**,**K**).

**Figure 6 ijms-22-06760-f006:**
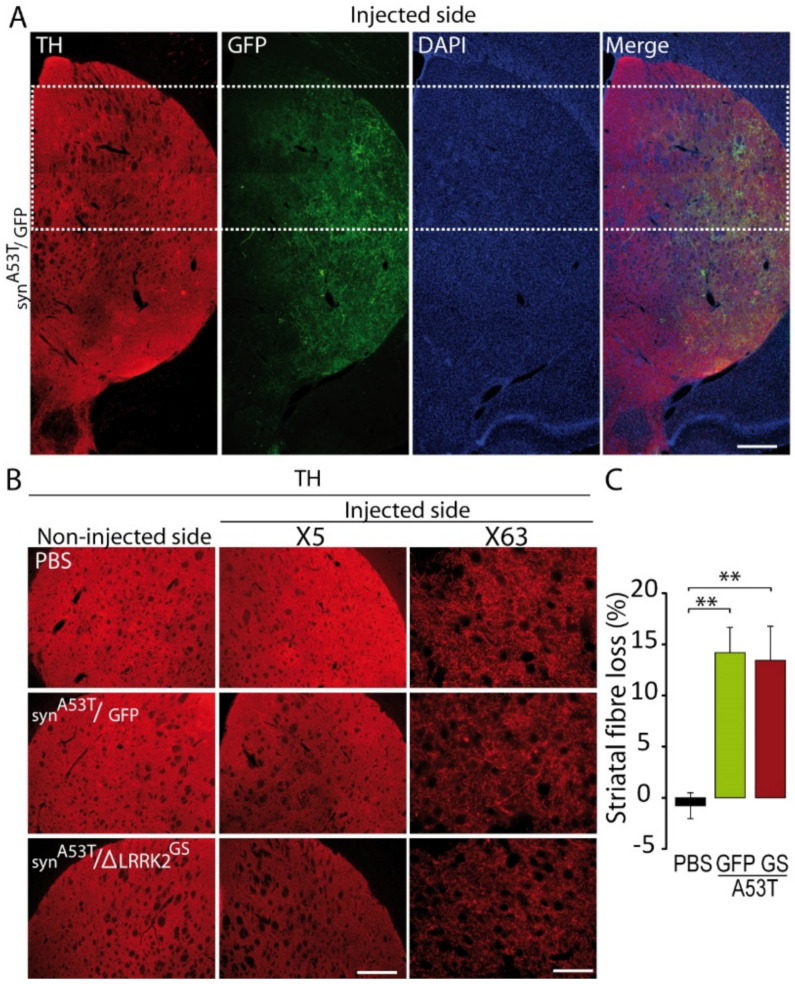
Tyrosine hydroxylase (TH) levels in the striatum of rats injected with AAV-α-syn^A53T^ with AAV-GFP or ΔLRRK2^G2019S^ in the right SNpc. (**A**) Photomicrographs at low magnification showing immunofluorescence for TH (red) and GFP (green) in the striatum at 15 weeks PI. Scale bar: 1000 µm. (**B**) Photomicrographs at two different magnifications (5× and 63×) showing TH-related immunofluorescence in the striatum of rats injected with PBS or AAV-α-syn^A53T^ with AAV-GFP, as a control, or ΔLRRK2^G2019S^ (GS). Scale bar in B: 5×, 1000 µm, 63×, 100 µm. (**C**) Quantification of fluorescence in the striatum. Quantification was performed at 5× magnification. Results are expressed as the means ± SEM. N = 10 animals/group. ANOVA and PLSD post hoc test. ** *p* < 0.01.

**Figure 7 ijms-22-06760-f007:**
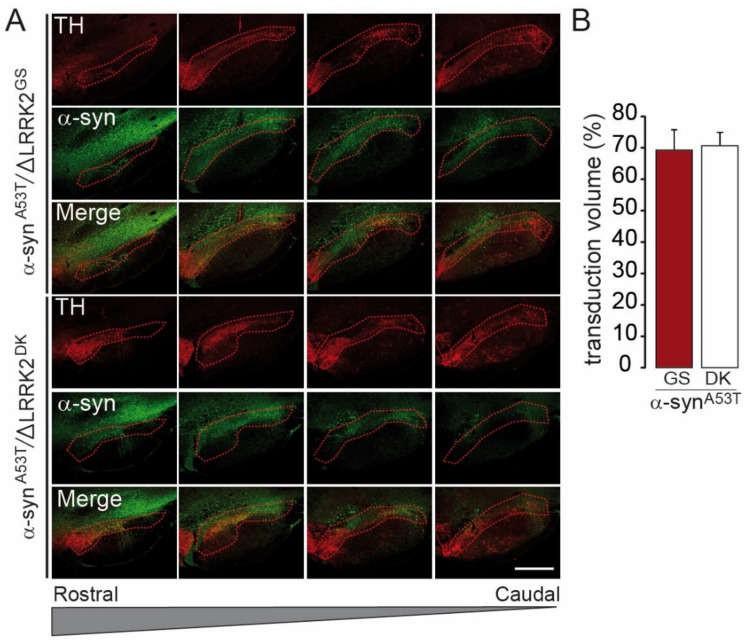
Measurement of the SNpc volume transduced by AAV-ΔLRRK2^G2019S^ and the dead kinase form ΔLRRK2^G2019S/D1994A^. (**A**) Confocal images to delineate the SNpc based on TH staining (in red), reported in the green channel, corresponding to the α-syn immunofluorescence when co-expressed with ΔLRRK2^G2019S^ (ΔLRRK2^GS^) or ΔLRRK2^DK^, the dead kinase form ΔLRRK2^G2019S/D1994A^. Scale bar: 1000 µm. (**B**) Quantification of the fraction (%) of the SNpc expressing α-syn protein after co-transduction with ΔLRRK2^G2019S^ or ΔLRRK2^G2019S/D1994A^. Results are expressed as the means ± SEM. N = 8 animals/group. No statistical difference, Student’s *t*-test.

**Figure 8 ijms-22-06760-f008:**
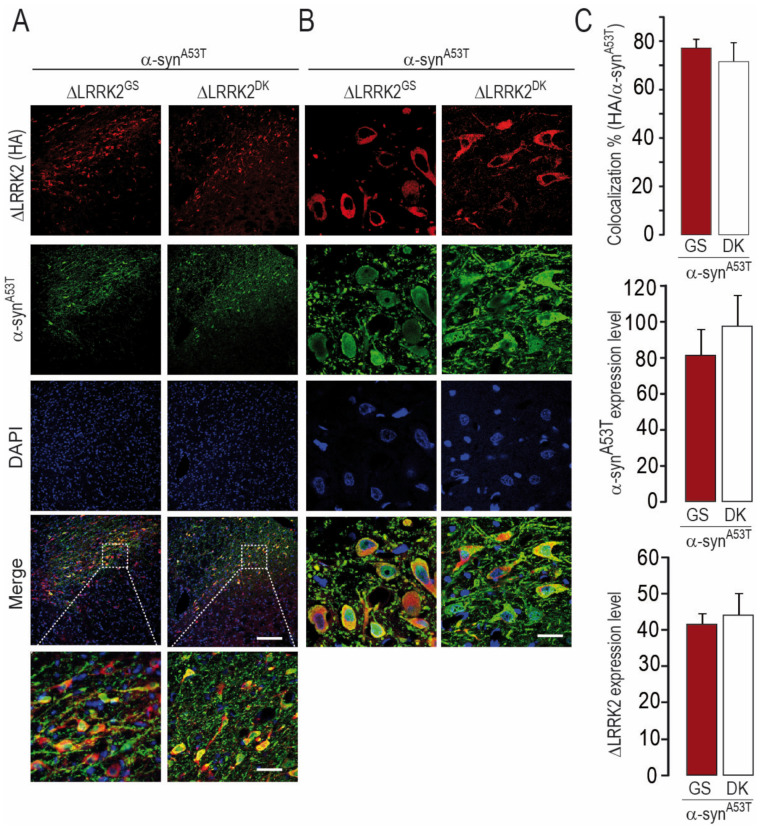
Co-localization and expression of ΔLRRK2 and α-syn^A53T^ 6 weeks after the co-injection of AAV-α-syn^A53T^ with either AAV-ΔLRRK2^G2019S^ or AAV-ΔLRRK2^G2019S/D1994A^. (**A**,**B**) Photomicrographs showing the results of immuno-fluorescence detection of ΔLRRK2^G2019S^ (ΔLRRK2^GS^) or ΔLRRK2^G2019S/D1994A^ (ΔLRRK2^DK^–red channel) and α-syn (green channel). Left and right images were obtained at low (**A**) and high (**B**) magnification, respectively. Note that most neurons express both transgenes. Scale bars: 200 µm for the top images, 50 µm for the bottom images in (**A**), and 10 µm for the bottom images in (**B**). (**C**) Quantification of the percentage of co-localization (upper histogram), α-syn fluorescence (middle histogram), and ΔLRRK2 fluorescence (bottom histogram) based on the analysis of 20 cells per animal, 180 cells in total. Results are expressed as the means ± SEM. N = 4–5 animals/group. No statistical difference between groups, Student’s *t*-test.

**Figure 9 ijms-22-06760-f009:**
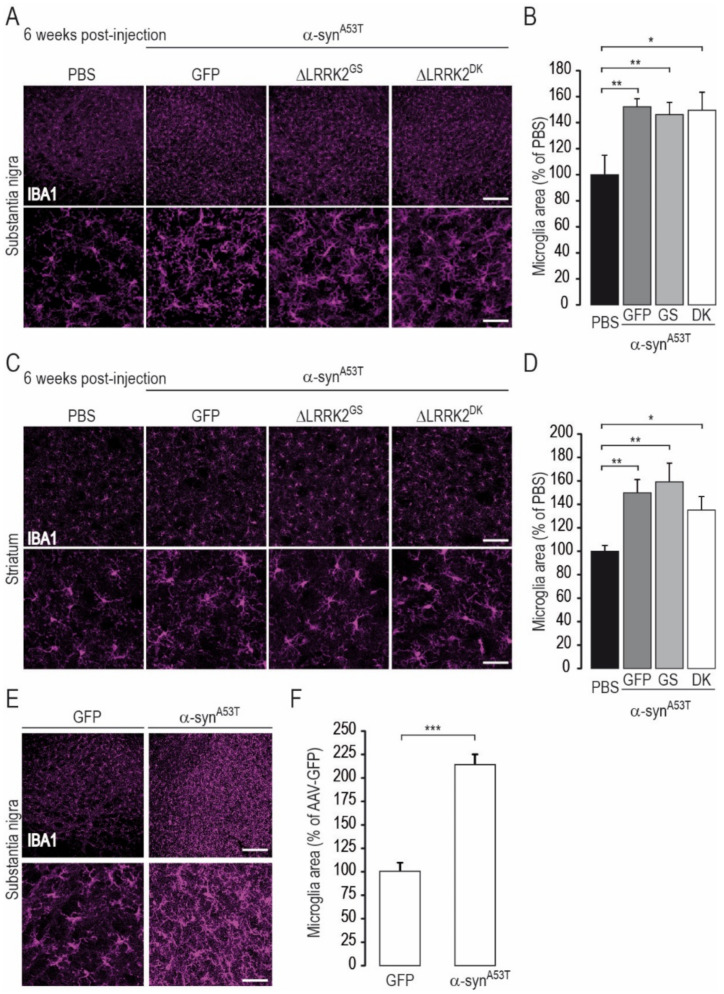
Microglial activation induced by α-syn^A53T^ is not modified by overexpression of ΔLRRK2 fragments. Histological evaluation was performed six weeks after the injection of PBS or AAV-α-syn^A53T^ alone or AAV-α-syn^A53T^ mixed with either AAV-GFP, ΔLRRK2^G2019S^ (ΔLRRK2^GS^), or the dead kinase form ΔLRRK2^G2019S/D1994A^ (ΔLRRK2^DK^). Cells positive for IBA1 were detected by immunofluorescence and confocal microscopy and their cross-sectional area was determined by image analysis. (**A**) Photomicrographs of rat brain sections labeled for IBA1 immunoreactivity in the SNpc at low (upper images) and high (lower images) magnification in the various groups. (**B**) Quantification of the mean cross-sectional area of IBA1-positive cells. (**C**) Low (upper images) and high (lower images) magnification photomicrographs of rat brain sections labeled for IBA1 immunoreactivity in the striatum of rats in which the SNpc was injected with PBS or AAV-α-syn^A53T^ with AAV-encoding ΔLRRK2 constructs. (**D**) Quantification of the mean cross-sectional area of IBA1-positive cells. (**E**) Low (upper images) and high (lower images) magnification photomicrographs of rat brain sections labeled for IBA1 immunoreactivity in the SNpc of rats injected with AAV-GFP or AAV-α-syn^A53T^ alone. (**F**) Quantification of the mean cross-sectional area of IBA1-positive cells. Results are expressed as the mean percentage ± SEM of the staining of the control group (PBS in (**A**–**D**), AAV-GFP in (**E**,**F**)). N = 8 animals/group in (**B**–**D**) and 5 to 7 animals/group in (**F**). In (**B**), ANOVA and PLSD post hoc test B (SNpc), and Kruskal–Wallis and Mann–Whitney tests in (**D**) (striatum). (**F**), Unpaired Student’s *t*-test. * *p* < 0.05, ** *p* < 0.01, *** *p* < 0.0001. Scale bar: low magnification, 200 µm; high magnification, 50 µm.

## Data Availability

Raw data might be provided on demand.

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
