# Peer review of "The C-Terminal Domain of LRRK2 with the G2019S Substitution Increases Mutant A53T α-Synuclein Toxicity in Dopaminergic Neurons In Vivo"

_ijms, 2021, doi:10.3390/ijms22136760_

Round 1

Reviewer 1 Report

This is an interesting in vivo study of LRRK2 c-terminal fragment on syn-A53T-linked neurodegeneration in rats.  The results showed that Co-injection of AAV-synA53T with AAV-LRRK2G2019S induced DA neurons degeneration that was significantly higher than that induced by AAV-synA53T alone, or with AAV-31 LRRK2DK. However, there is normal motor performance in syn53t/GS rats compared with syn53T rats.  There is also no change in gliosis in syn53t/GS compared with synA53T rats. The findings are interesting. However, it raises some confusion or contradiction below.  The further experiments can validate or clarify the current findings.

  1. Fig 1 showed that rats injected with AAV-LRRK2 fragment did not cause any TH neurons loss at 15 weeks of age. In the discussion (page 14) “We previously showed that overexpression of the LRRK2G2019S fragment using AAVs triggers neurodegeneration of DA neurons six months PI, whereas the LRRK2WT fragment, expressed at similarly high levels, was devoid of obvious neurotoxicity [32]”. Why authors do not observe in 6 months for these co-injection rats and see how the results are? This may further validate authors’ current conclusion with late time point study (more clinical phenotypes).

2.  In Fig 4A motor test results, it appears that GS-LRRK2 rescues the motor defects in syn153T/GS-LRRK2 con-injected mice, which raises a contradiction with neuron loss data. Fig 5 showed DA neurons loss more in synA53t/GS con-injected rats.  Authors provide some discussion of about this and stated that this could due to DA compensation. Thus, measure the dopamine in SN is necessary to clarify and understand these contradicted findings.

Author Response

We addressed all the points raised by reviewers.

Detailed answers are in the attached documents.

Reviewer 2 Report

This manuscript tried to investigate the interplay of LRRK2 and alpha-synuclein (aSyn) in the dopaminergic neurons. The authors chose to use an AAV-based approach with the C-terminal domain of LRRK2 with G2019S mutation and mutant A53T aSyn. It tested a fundamental hypothesis that whether LRRK2 G2019S pre-deposited aSyn vulnerability in the dopamine neurons. The results supported, the authors argued, a "pro-toxic" effect of LRRK2 G2019S on aSyn A53T in vivo.

The study was carefully conducted with baseline controls of LRRK2 G2019S/DK and aSyn A53T virus injection alone. It compared the TH expression, p-aSyn, and microglia activation at six weeks and 15 weeks PI. However, there were unexpected behavioral results with conflicting neurological assessments.

Major concerns:

  1. The expression level of aSyn A53T was not thoroughly evaluated or compared. The authors correctly pointed out the difficulties in comparing human aSyn with rat aSyn levels (line 445-446). However, using only fluorescence intensities in 20 cells (line 671-674) was not sufficient to compare aSyn levels across different groups, especially A53T+GS vs. A53T+GFP (Figure 5B) or A53T+GS vs. A53T+DK (Figure 9A). It is necessary to rule out the possibility that the observed difference of TH+ cells was not due to the difference in A53T expression levels.
  2. Conflicting data regarding p-aSyn level in the striatum at two-time points. First, disproportional p-aSyn level for A53T+DS vs. A53T+GFP group at 15 weeks PI (Figure 5F, 5% vs. 15%), yet the same levels of striatal fiber loss (Figure 6C, ~13%) at the same time point. The authors stated that the lower level of p-aSyn was due to severe loss of TH+ neurons in SNpc (line 502-504). However, if it were true, there should be a significant loss of TH immunofluorescence in the striatum in the A53T+GS group. Second, there was reported ~16% of p-aSyn in the striatum for A53T+GS group at six weeks PI (Figure 9F), much higher than the 5% at 15 week PI. Assuming the decrease of p-aSyn immunoactivity in the striatum was solely caused by the loss of p-aSyn+ neurons in SNpc, as the authors suggested and no p-aSyn addition in the striatum from 6 to 15 weeks PI (unlikely), then there should be at least two-thirds of those p-aSyn+ neurons (Figure 9D, ~12000) dead during this period, which should be left with <4000 p-aSyn+ neurons at 15 weeks PI. Nevertheless, over 6000 p-aSyn+ neurons were reported in Figure 5D. In addition, the p-aSyn level in the striatum was evaluated as the percentage of the field view (line 250-251), which should reflect the absolute p-aSyn burdens in the terminals, independent of the neuronal loss in the SNpc.
  3. Unwarranted p-aSyn+ / TH+ neurons ratio. The authors use the ratio to explain the significantly lower p-aSyn positive cells in the A53T+GS group. The ratios were bigger than one for A53T+GFP and A53T+PBS group at 15 weeks PI, which could not indicate "the presence of p-aSyn in surviving cell soma (line 259-262)." In addition, there could be TH negative neurons with positive p-aSyn and TH positive neurons with negative p-aSyn in SNpc. This ratio has no physiological meaning to reflect the p-aSyn levels in surviving or degenerating neurons.
  4. Over-reliance on the cylinder test as the behavioral benchmark. The asymmetry scores at six weeks PI for A53T alone were significantly lower than the baseline (Figure 2A), yet none of the co-injected groups (A53T+GFP/GS/PBS) demonstrated any significant deficiency at six weeks PI (Figure 4A). As the authors discussed, the behavioral results at 15 weeks PI were "counter-intuitive." The authors should apply additional behavioral tests to investigate the discrepancy between the phenotype with the loss of dopaminergic neurons. Also, the amphetamine test only reported the total rotation rather than total distance traveled, and no open-field tests were conducted to support the conclusion of "higher release of DA" in the A53T+GS group (line 397-398).
  5. The authors chose different time points to investigate different parts of the hypothesis. There was no DK group at 15 week PI and no A53T+PBS at six weeks PI. It would be much easier to understand the results if the figures were combined to form two time points (6 and 15 weeks PI) with six groups: PBS, A53T alone, A53T+PBS, A53T+GFP, A53T+GS, and A53T+DK.

Minor concerns:

  1. Line 132. The authors might want to explain the reason behind the choice of a slightly larger fragment (aa 1330-2527 vs. 1283-2527).
  2. Figure 1B. Reference or discussion needed for the notable behavioral deficiency of the LRRK2 DK group at 15 weeks. The authors argued the "cell dysfunction," yet the asymmetry level was comparable to the A53T at 15 weeks PI, which suffered 30% loss of TH in SNpc.
  3. Figure 3. The delineation of SNpc with TH staining at 15 weeks PI underestimated the total area of SNpc. At 15 weeks, at least 30% of TH lost is indicated by Figure 2C. Thus, it is possible that part of the SNpc infected by A35T turned the TH off and did not transduce LRRK2.
  4. Figure 5C and 9C. The images showed intense p-aSyn stainings in the uninjected side of the SN pars reticulata. Endogenous p-aSyn? Comparable to the injected side?
  5. Figure 7. Measurement of SNpc volume transduced by LRRK2 GS and Dk were underestimated by delineating the SNpc based on TH staining for the same reason in # 3 (with less TH loss at six weeks, yet exact percentage TH loss unreported).
  6. Figure 8. A53T and LRRK2 expression levels were not adequately quantified. The fluorescence levels were often incomparable between slices, which, more importantly, were not equal to the protein expression levels in the cells or tissues.
  7. Figure 9D. More or at least equal number of p-aSyn+ cells compared with the total number of dopaminergic neurons in SNpc (~12000 in PBS group in Figure 1). So, nearly all TH positive and negative neurons contained p-aSyn at six weeks in the A53T+GFP group?
  8. Figure 9D. The ratios were all bigger than one, indicating more p-aSyn+ neurons than TH+ neurons at six weeks PI. The authors did not discuss the meaning of those ratios at six weeks PI or the reduction of the ratios from 1.5 to 1.2 at 15 weeks PI (Figure 5D).
  9. Line 319-320. No asymmetry data in Figure 9.
  10. Figure 10. The control levels of IBa1, such as A53T and GFP alone, were not compared with PBS or co-injection groups. IBa1 levels of background groups, GS and DK alone, were not reported.
  11. Multiple places. The TH loss is not equal to neuronal death. Turning off TH expression is not necessarily a marker for cell loss, although lack of TH staining is a clear sign of degeneration.

Author Response

We addressed all the points raised by the reviewers.

Detailed answers are in the attached document.

Round 2

Reviewer 2 Report

The revised manuscript has clarified most of the concerns in the previous review. In addition, the authors provided a detailed and convincing cover letter that addressed all the listed points. The revision has improved both the scientific soundness and quality of the presentation significantly. 

Only a few comments for the authors:

  1. The original review contains no objection to the fluorescence intensities analysis method for quantifying A53T expression levels. The question was whether that analysis is sufficient to address such an important issue. One might argue that the qPCR might be the most sensitive and reliable method for this type of analysis. However, the fluorescence analysis contains spatial information that no qPCR or Western blots can provide. Although the results most likely supported the authors' conclusion, the fundamental issue was in the cells infected with both viruses, whether the co-expression affected the expression level of A53T, assuming the effect of G2019S is restricted in a cell-autonomous fashion. 
  2. The role of p-aSyn in pathology is the core question here. Recently, some studies argue that phosphorylation promotes aSyn degradation as a protective mechanism rather than a degeneration biomarker. The molecular mechanism behind the high level of p-aSyn at 6-week PI and the sharp decrease from 6 to 15 weeks is requesting further investigation. 
  3. The cylinder test is a widely used test for unilateral DA loss. However, the test highly relies on the animal's willingness to participate (put a hand on the cylinder when "standing" during the trial spontaneously.) So the test contained some levels of variability. When co-tested with other tests such as the forelimb placement test and bracing test, the results demonstrated the lowest sensitivity towards the behavioral asymmetry in our own hands.